# NFIG: Multi-Scale Autoregressive Image Generation via Frequency Ordering

**Zhihao Huang**[1,2] **Xi Qiu**[2] **Yukuo Ma**[2,4] **Yifu Zhou**[1,2] **Junjie Chen**[2]
**Hongyuan Zhang**[2,3,*] **Chi Zhang**[2,*] **Xuelong Li**[2,*]

[1] Northwest Polytechnical University
[2] TeleAI, China Telecom    [3] University of Hong Kong    [4] Beihang University

huangzhihao@mail.nwpu.edu.cn, hyzhang98@gmail.com,
zhangc120@chinatelecom.cn, xuelong_li@ieee.org

## Abstract

Autoregressive models have achieved significant success in image generation. However, unlike the inherent hierarchical structure of image information in the spectral domain, standard autoregressive methods typically generate pixels sequentially in a fixed spatial order. To better leverage this spectral hierarchy, we introduce **N**ext-**F**requency **I**mage **G**eneration (**NFIG**). NFIG is a novel framework that decomposes the image generation process into multiple frequency-guided stages. NFIG aligns the generation process with the natural image structure. It does this by first generating low-frequency components, which efficiently capture global structure with significantly fewer tokens, and then progressively adding higher-frequency details. This frequency-aware paradigm offers substantial advantages: it not only improves the quality of generated images but crucially reduces inference cost by efficiently establishing global structure early on. Extensive experiments on the ImageNet-256 benchmark validate NFIG's effectiveness, demonstrating superior performance (FID: 2.81) and a notable $1.25\times$ speedup compared to the strong baseline VAR-d20. The source code is available at https://github.com/Pride-Huang/NFIG.

## 1 Introduction

[2]

The synthesis of images has emerged as a fundamental challenge in computer vision [1, 2, 3, 4, 5, 6, 7, 8, 9, 10, 11, 12]. Rapid progress in this field has been propelled by deep generative models, such as autoregressive models (AR) [13], Generative Adversarial Networks (GANs), and diffusion models (SD) [14].

Despite remarkable advances in existing methods, AR models for image generation still face several fundamental challenges. On the one hand, due to their inherently local and sequential nature, most current AR models struggle to effectively capture long-range dependencies and global structure [15]. For example, PixelCNN [9] generates an image by predicting each pixel in a raster scanning sequence, which neglects the global image structure and relationships with distant elements. On the other hand, the generation process is computationally intensive and time-consuming, as AR models always generate pixels or patches sequentially in a predetermined order, with each new element requiring the computation of conditional probabilities based on all previously generated content [16, 17]. For instance, ViTVQ [18] requires more than 6 seconds to generate a $256 \times 256$ image over

---

*Corresponding authors
[2]Completed during internship at TeleAI.

39th Conference on Neural Information Processing Systems (NeurIPS 2025).

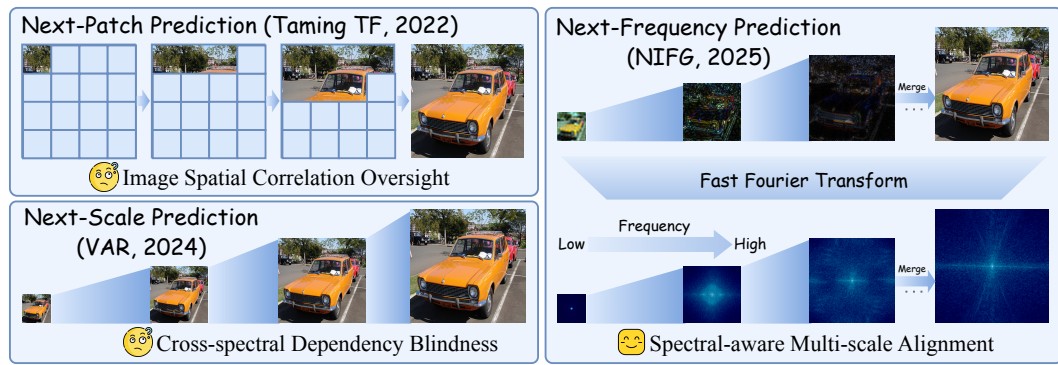

Figure 1: Illustration of three autoregressive image generation frameworks. The figure demonstrates three prediction approaches: Next-Patch Prediction (patch-based progression), Next-Scale Prediction (coarse-to-fine resolution generation), and Next-Frequency Prediction (NFIG), which performs image generation by progressively predicting and synthesizing frequency components from low to high, resulting in a coarse-to-fine spatial reconstruction.

1024 steps, making it impractical for real-time applications. Most importantly, AR models face a fundamental challenge in defining a meaningful autoregressive sequence. Traditional AR models using raster scanning or predefined arbitrary orders fail to reflect the natural hierarchical structure and dependencies in images[19]. This improper sequence design makes it difficult for models to capture the true causal relationships between different image components, ultimately affecting the coherence and visual quality of generated outputs.

To address these limitations, recent works have explored incorporating various improvements into the generation process. For example, Taming Transformer [20] partially addresses long-range dependency challenges through its discrete latent space and transformer architecture, but still suffers from computational inefficiency. Fast PixelCNN++ [21] speeds up generation in convolutional autoregressive models by caching hidden states to avoid redundant computation, achieving up to 183× speedups, yet it doesn't fundamentally change the autoregressive sequence design. VAR [19] leverages the Laplacian Pyramid as a prior to guide autoregressive image generation across different resolutions, achieving improved generation quality with reduced computational overhead. However, these methods do not fully exploit the potential of natural priors inherent in raw images to guide the generation process and improve the efficiency of AR models.

In fact, ***the natural structure of images follows a hierarchical frequency distribution—low frequencies encode global structures while high frequencies contain local details.*** This organization suggests an efficient autoregressive generation sequence from low to high frequencies, aligning with visual information's natural structure. Since low-frequency components require fewer tokens to represent, this approach enhances computational efficiency. Similar frequency-progressive principles have proven effective in diffusion models, which build from low-frequency foundations before adding higher-frequency details [22].

Motivated by this insight, we propose a Next-Frequency Image Generation (NFIG) framework for AR models that: (1) first generates a low-frequency image with few tokens to capture global structure; (2) then progressively adds higher-frequency components conditioned on the low-frequency foundation. This process has been shown in Figure 1. Grounded in information theory, this approach efficiently represents information across the frequency spectrum using our Frequency-guided Residual-quantized VAE (FR-VAE).

Key contributions of the NFIG framework include:

- We introduce a Next-Frequency Image Generation (NFIG) framework that incorporates frequency analysis into AR image generation. To our knowledge, this work is the first to guide autoregressive generation using the image's frequency spectrum, associating low frequencies with lower resolutions and high frequencies with higher resolutions;

- To demonstrate the feasibility of the NFIG paradigm, we design a Frequency-guided Residual-quantized VAE as our image tokenizer. FR-VAE separates low and high-frequency components in the representation learning process, with low frequencies encoding global structure and high

frequencies preserving local details. Experiments show FR-VAE achieves a reconstruction FID of 0.85, validating its image content preservation capability;

• Through extensive experimentation, we show that our approach achieves state-of-the-art image generation quality, evidenced by an FID of 2.81 from a relatively small model. This improvement paves the way for more effective and efficient AR image generation models, making them more practical for real-world applications.

Table 1: Main Notation Table

| Symbol | Meaning | Dimension |
|--------|---------|-----------|
| $x$ | Input image | $H \times W \times 3$ |
| $\hat{x}$ | Reconstructed image | $H \times W \times 3$ |
| $f$ | Feature map from VAE encoder | $H' \times W' \times C$ |
| $M_i$ | The $i$-th frequency selection mask | $H' \times W' \times C$ |
| $\hat{f}_i$ | Set of $i$-th frequency component feature maps | $H' \times W' \times C$ |
| $v_i$ | Scaled feature map for $i$-th frequency component | $h_i \times w_i \times C$ |
| $v_i^q$ | Quantized representation for $i$-th frequency component | $h_i \times w_i \times C$ |
| $R_i$ | Cumulative signal residual through level $i$ | $H' \times W' \times C$ |
| $Z$ | The learnable codebook of FR-VAE | $K \times C$ |
| $F_i$ | The $i$-th frequency band | $N/A$ |

## 2 Related Work

**Autoregressive Image Generation** Autoregressive image generation has demonstrated remarkable capabilities in producing high-quality images by modeling the joint distribution of image tokens as a product of conditionals[23]. PixelCNN [9] generates images sequentially, processing pixels one by one (typically top-left to bottom-right). It employs masked convolutions so that the generation of each pixel depends solely on the pixels already generated. Taming Transformer [20] introduces an autoregressive approach that generates high-resolution images by predicting the next latent patch token in a discrete compressed space learned through vector quantization. Emu3 [24] patchifies an image into a series of tokens and generates images by predicting tokens in a raster-scan manner. VAR [19] incorporates the prior knowledge of Laplacian Pyramid into transformer architecture and generates images in a next-resolution manner. MAR [25] improves the quality of generated images by replacing discrete tokens with continuous features, recognizing that autoregressive models primarily need per-token probability distributions. FAR [26] attempts to enhance MAR performance through frequency-based approaches, yet lacks critical insights into the distinctive information characteristics across different frequency bands. Infinity [25] combines autoregressive modeling with Bit-wise Modeling to enhance visual details in high-resolution image synthesis. ImageFolder [27] utilizes folded image tokens to generate high-quality images in a next-scale prediction manner, achieving superior performance. These approaches collectively demonstrate the evolution of autoregressive image generation techniques, progressing from pixel-level prediction to more sophisticated methods involving latent spaces, hierarchical structures, and physical priors. Despite their differences in implementation, all these methods share the fundamental autoregressive principle of sequentially generating image elements conditioned on previously generated content.

**Image Tokenizer** Image tokenizers, which transform continuous image data into discrete representations, have become a critical component in modern image generation systems. VQ-VAE [28] introduces vector quantization into VAE, reducing the pressure on the downstream generative model by transforming the continuous latent space into a discrete one. VQ-VAE-2 [29] extends this idea with a hierarchical framework and multi-scale codebooks, where top-level codes capture global structure, and bottom-level codes model local details, enabling higher-quality reconstruction and generation at increased resolutions. However, VQ-VAE-based methods often suffer from codebook collapse, where only a few codebook entries are effectively used. FSQ [30] attempts to address this issue by utilizing finite scalar quantization to learn the codebook, but it does not learn a meaningful feature of images. RQ-VAE [31] employs residual quantization with a shared codebook to enhance reconstructed image quality. XQGAN [32] introduces feature product decomposition and a residual quantizer to enhance VQ-VAE's performance, leading to improved image generation results. While these approaches have

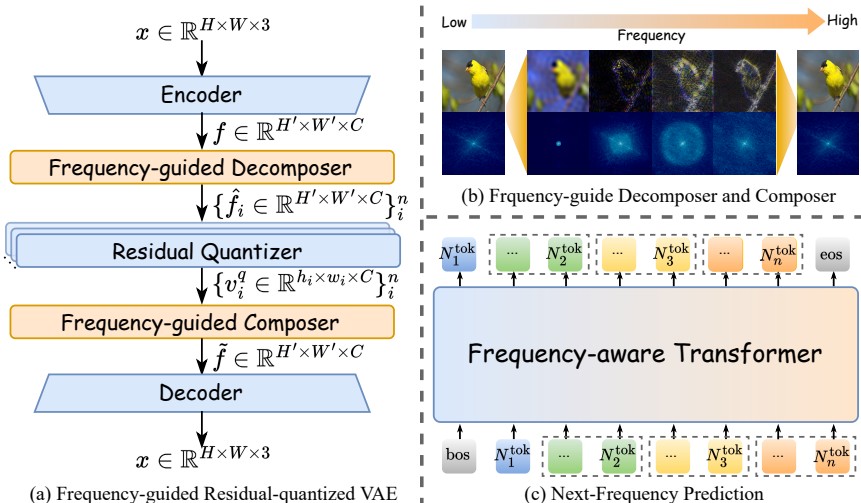

(a) Frequency-guided Residual-quantized VAE

(b) Frquency-guide Decomposer and Composer

(c) Next-Frequency Prediction

Figure 2: Overview of the Next-Frequency Image Generation (NFIG) Framework: (a) The Frequency-guided Residual-Quantization VAE encodes images into and decodes from frequency-guided residual quantized representations; (b) The image is decomposed into frequency components (low to high) and reconstructed progressively by merging these components for a coarse-to-fine process; (c) Next-Frequency Prediction model employs a frequency-aware Transformer to auto-regressively generate token sequences, with each block with same color representing a specific frequency band, enabling sequential image synthesis from low to high frequencies. $N_i^{tok} = h_i w_i$ is the number of image tokens used for the $i_{th}$ frequency band.

made significant progress in image tokenization, they often neglect the inherent multi-scale structure of natural images, which is crucial for efficient and effective representation learning.

# 3 Next-Frequency Image Generation

To provide a comprehensive understanding of our NFIG methodology, this section delves into its intricate architectural structure. The essential operational sequence, illustrating the flow and interaction of the system's key components, is clearly visualized in Figure 2. The details of loss function have been listed in Appendix B.1.

## 3.1 Frequency-guided Residual-quantized VAE

The workflow of Frequency-guided Residual-quantized VAE has been shown in Figure 2 (a). To generate images in a frequency-aware manner, we propose a Frequency-guided Residual-quantized VAE (FR-VAE) with VQ-GAN framework. The key idea is to represent lower-frequency signals with fewer tokens and higher-frequency components with more tokens.

### 3.1.1 Frequency-guided Reconstruction

As illustrated in Figure 2 (b), raw images can be decomposed into components across different frequency bands: low frequencies encode the global structure, while high frequencies retain fine details. Utilizing the Frequency-guided Decomposer and Composer, these components can be recombined without loss, ensuring a complete and accurate visual representation.

**Frequency-guided Decomposer.** Given a image $x \in R^{H \times W \times 3}$ and a encoder $E(\cdot)$, there is image latent feature $f = E(x)$ and $f \in \mathbb{R}^{H' \times W' \times C}$. FR-VAE decomposes $f$ into several component with

different frequency by Frequency-guided Decomposer via Fast Fourier Transform (FFT):

$$\hat{f}_i = \mathcal{F}^{-1}(\mathcal{F}(f) \odot M_i), \forall i \in \{1, \cdots, n\}. \tag{1}$$

Here, $\odot$ represents the element-wise product, $\mathcal{F}$ signifies the FFT operation, $\mathcal{F}^{-1}$ indicates the inverse FFT, and $M_i$ is the $i$-th frequency mask used to select the desired frequency range, $n$ is the total number of frequency masks, $\hat{f}_i$ is the component corresponding to $M_i$.

**Frequency-guided Composer.** Frequency-guided Composer reconstructs the raw image by interpolating different frequency components to a uniform size and merging them into a single image, as illustrated in Figure 2 (b).

$$\tilde{f} = \sum_{i=1}^{n} \mathcal{I}(\hat{f}_i, H', W'), \tag{2}$$

where $\mathcal{I}(\cdot, H', W')$ is the interpolation function, which enables the Frequency-guided Composer to process images of different frequencies at varying resolutions.

### 3.1.2 Frequency-guided Residual-quantization

To efficiently represent images with minimal tokens, we implement a frequency-guided residual quantization approach that addresses information loss during downsampling. Our method progressively captures different frequency components of an image through a residual learning scheme.

**Residual Token Extraction.** Given a sequence of feature maps with different dimensions $\{(h_1, w_1), \cdots, (h_n, w_n)\}$, where $h_i \geq h_j$ and $w_i \geq w_j$ if $i \geq j$, and $h_n = H'$ and $w_n = W'$, we supervise the learning process using accumulated signals from the lowest frequency to the current frequency band.

The residual $R_i \in \mathbb{R}^{H' \times W' \times C}$ and representation $v_i \in \mathbb{R}^{h_i \times w_i \times C}$ of the $i$-th frequency component can be computed as follows:

$$R_i = \begin{cases} \hat{f}_i - \mathcal{I}(v_i, H', W')), & i = 0 \\ R_{i-1} + (\hat{f}_i - \mathcal{I}(v_i, H', W'))), & i \geq 1 \end{cases}, \tag{3}$$

$$v_i = \begin{cases} \arg\min_{v_i} \|\hat{f}_i - \mathcal{I}(v_i, H', W')\|^2, & i = 0 \\ \arg\min_{v_i} \|(R_{i-1} + \hat{f}_i) - \mathcal{I}(v_i, H', W')\|^2, & i \geq 1 \end{cases}, \tag{4}$$

where $\mathcal{I}(v_i, H', W')$ is the interpolation function that upsamples $v_i$ to the original feature map size, and $R_i$ represents the difference between the accumulated frequency components up to the $i$-th level and the learnable features.

**Vector Quantization.** In general, autoregressive models utilize the discrete tokens to generate a image. To achieve this goal, we take a simple vector quantization to transform the continuous token into discrete tokens.

We define a quantizer $Q$ with a learnable codebook $Z \in \mathbb{R}^{K \times C}$ containing $K$ code vectors. Using this codebook, the quantizer $Q$ transforms a continuous feature map $v_i \in \mathbb{R}^{h_i \times w_i \times C}$ into a set of discrete tokens $\{t_i^{(1,1)}, t_i^{(1,2)}, \cdots, t_i^{(h_i, w_i)}\}$, where each token $t_i^{(j,k)}$ has the corresponds to a vector $z_i^{(j,k)} \in \mathbb{R}^C$. The process of finding the optimal code representation involves:

$$t^{(j,k)} = \text{lookup}(Z, \arg\min_{z_i^{(j,k)} \in Z} \|z_i^{(j,k)} - v_i^{(j,k)}\|_2). \tag{5}$$

From $v_i$, the quantized feature map $v_i^q \in \mathbb{R}^{h_i \times w_i \times C}$ and a set of discrete tokens $\{t_i^{(j,k)}\}$ are obtained through quantization using the codebook $Z$. Here, $\text{lookup}(Z, x)$ is a function that finds the index of the closest entry to $x$ in codebook $Z$.

### 3.2 Autoregressive Image Generation

To generate images progressively from low to high frequency components, we implement a decoder-only transformer framework and block-wise causal attention [19].

**Next-Frequency Image Prediction**    Unlike conventional autoregressive image generation models that employ a "token-by-token prediction" strategy, which often neglects spatial relationships and inherent image structure. NFIG adopts a "Coarse-to-Fine Generation" approach, first synthesizes the low-frequency components of an image, then iteratively incorporates higher-frequency details, progressively refining the generated output at each step, as shown in Figure 2 (c). The generation process for next-frequency prediction is given by the autoregressive factorization:

$$p(T_1, T_2, \cdots, T_n) = \prod_{i=1}^{n} p(T_i | T_1, T_2, \cdots, T_{i-1}) \tag{6}$$

where $T_i \in [K]^{h_i \times w_i}$ is the matrix of code indices for the $i$-th frequency component, and the set $[K] = \{1, 2, \ldots, K\}$ represents all available index values.

**Frequency Band Division Strategy**    We treat the lower frequency components as the foundation for generating $T_i$, represented by $\{T_1, T_2, \cdots, T_{i-1}\}$. According to information theory principles, lower-frequency signals contain less information and require fewer tokens, while higher-frequency components carry more detailed information and need more tokens for accurate representation.

Consequently, we establish an increasing scale sequence $\{(h_1, w_1), (h_2, w_2), \cdots, (h_n, w_n)\}$ for components with increasing frequency bands $\{F_1, F_2, \cdots, F_n\} = \{[0, \sigma_1), [\sigma_1, \sigma_2), \cdots, [\sigma_{n-1}, \sigma_n]\}$. Here, $\sigma_{max}$ denotes the maximum frequency of the entire image feature map $f$, with $\sigma_n = \sigma_{max}$. We divide the frequency bands based on their corresponding resolution as:

$$\sigma_i = \sigma_{i-1} + \frac{h_i \cdot w_i}{\sum_{j=1}^{n} h_j \cdot w_j} \times \sigma_{max}. \tag{7}$$

This frequency-guided progressive approach allows our model to capture and prioritize salient components at each stage. The method improves both computational efficiency and image quality by explicitly modeling the multi-scale frequency structure inherent in natural images.

## 4    Experiment

This section details our experimental methodology, covering datasets, evaluation metrics, comparison baselines, and implementation specifics. We then evaluate NFIG against state-of-the-art approaches on image generation benchmarks. Subsequently, ablation studies and motivation verification experiments are conducted to analyze the impact of different components and validate design decisions.

### 4.1    Experimental Settings

**Dataset**. For the purpose of our experiments, we use the ILSVRC 2012 subset of ImageNet [33], which comprises a total of 1.2 million training images, 50k validation images, and 100k test images. This subset focuses on 1k object categories, with each category having approximately 1.2k training images, 50 validation images, and 100 test images.

**Evaluation Metrics**. We adopt four metrics for quantitative evaluation: Fréchet Inception Distance (FID) which measures distribution similarity between generated and real images, Inception Score (IS) which assesses quality and diversity, and Precision (Pre) and Recall (Rec) which evaluate sample fidelity and diversity coverage respectively.

**Baselines**. Our method is benchmarked against several leading image generation techniques, including generative adversarial networks (GAN), diffusion models (Diff.), mask diffusion (Mask.), and autoregressive models (AR). These approaches have demonstrated strong performance on various image synthesis tasks and serve as robust comparators.

**Implementation Details**. Our model was implemented using the PyTorch framework [44] and trained on NVIDIA H100 graphics cards. To ensure the reproducibility of our experiments, our implementation is built upon open-source research code, while incorporating improvements specific to this study. For the image tokenizer, the FR-VAE incorporates a VQGAN architecture with a DINO discriminator. The image encoder is initialized with pretrained weights from DINOv2-base. Since VAR's image tokenizer training code is not open-source, we adopted XQGAN's implementation strategy. The frequency residual quantizer employs multiple scaling factors $[1, 2, 3, 4, 5, 6, 8, 10, 13, 16]$

Table 2: Performance on class-conditional ImageNet 256×256 for image generative model. rFID represents reconstruction FID , while gFID indicates generation FID. "↓" or "↑" indicate lower or higher values are better. "#Step": the number of model runs needed to generate an image. Wall-clock inference time relative to NFIG is reported. Models with the suffix "-re" used rejection sampling. †: taken from MaskGIT [34]. For comprehensive evaluation, we separately compare autoregressive (AR) and non-autoregressive (non-AR) models, with the **best metrics** highlighted in **bold**.

| Type | Model | rFID↓ | gFID↓ | IS↑ | Pre↑ | Rec↑ | #Para | #Step | Time |
|------|-------|-------|-------|-----|------|------|-------|-------|------|
| GAN | BigGAN [35] | - | 6.95 | 224.5 | **0.89** | 0.38 | 112M | 1 | – |
| GAN | GigaGAN [36] | - | 3.45 | 225.5 | 0.84 | 0.61 | 569M | 1 | – |
| GAN | StyleGan-XL [3] | - | 2.30 | 265.1 | 0.78 | 0.53 | 166M | 1 | 0.75 |
| Diff. | ADM [37] | - | 10.94 | 101.0 | 0.69 | **0.63** | 554M | 250 | 420 |
| Diff. | CDM [38] | - | 4.88 | 158.7 | – | – | – | 8100 | – |
| Diff. | LDM-4-G [39] | - | 3.60 | 247.7 | – | – | 400M | 250 | – |
| Diff. | DiT-L/2 [40] | **0.9** | 5.02 | 167.2 | 0.75 | 0.57 | 458M | 250 | 77.5 |
| Diff. | DiT-XL/2 [40] | **0.9** | 2.27 | 278.2 | 0.83 | 0.57 | 675M | 250 | 112.5 |
| Diff. | L-DiT-3B [41] | **0.9** | **2.10** | 304.4 | 0.82 | 0.60 | 3.0B | 250 | >112.5 |
| Diff. | L-DiT-7B [41] | **0.9** | 2.28 | **316.2** | 0.83 | 0.58 | 7.0B | 250 | >112.5 |
| Mask. | MaskGIT [34] | 2.28 | 6.18 | 182.1 | 0.80 | 0.51 | 227M | 8 | 1.25 |
| Mask. | RCG [42] | - | 3.49 | 215.5 | – | – | 502M | 20 | 4.75 |
| AR | VQVAE-2† [34] | 2.0 | 31.11 | 45.0 | 0.36 | 0.57 | 13.5B | 5120 | – |
| AR | VQGAN† [20] | 7.94 | 18.65 | 80.4 | 0.78 | 0.26 | 227M | 256 | 47.5 |
| AR | VQGAN [20] | 7.94 | 15.78 | 74.3 | – | – | 1.4B | 256 | 60 |
| AR | ViTVQ [18] | 1.28 | 4.17 | 175.1 | – | – | 1.7B | 1024 | >60 |
| AR | ViTVQ-re [18] | 1.28 | 3.04 | 227.4 | – | – | 1.7B | 1024 | >60 |
| AR | RQTran. [43] | 1.83 | 7.55 | 134.0 | – | – | 3.8B | 68 | 52.5 |
| AR | RQTran.-re [43] | 1.83 | 3.80 | 323.7 | – | – | 3.8B | 68 | 52.5 |
| AR | FAR-B [26] | - | 4.26 | 248.9 | 0.79 | 0.51 | 208M | 10 | - |
| AR | FAR-B [26] | - | 3.45 | 282.2 | 0.80 | 0.54 | 427M | 10 | - |
| AR | FAR-H [26] | - | 3.21 | 300.6 | 0.81 | 0.55 | 812M | 10 | - |
| AR | XQGAN-310M [32] | 0.78 | 2.96 | - | - | - | 310M | 10 | 1 |
| AR | VAR-d16 [19] | 0.9 | 3.55 | 274.4 | **0.84** | 0.51 | 310M | 10 | 1 |
| AR | VAR-d20 [19] | 0.9 | 2.95 | 302.6 | 0.83 | 0.56 | 600M | 10 | 1.25 |
| AR | NFIG(Ours) | **0.85** | **2.81** | **332.42** | 0.77 | **0.59** | 310M | 10 | 1 |

across different frequency bands, resulting in a vocabulary size of 680 tokens. The FR-VAE codebook size of 4096 was utilized. The image generator employs a VAR Transformer backbone with a depth of 16, enabling multi-scale image prediction. Optimization was performed using the Adam optimizer, setting the learning rate to $8 \times 10^{-5}$ and the batch size to 768. Training of the model ran for 350 epochs on the ImageNet dataset. For inference, we configured CFG to 4.5 and top_k to 990.

## 4.2 Main Results

Table 2 provides a detailed comparison of our approach against leading image generative models evaluated on ImageNet $256 \times 256$. The findings indicate that NFIG achieves superior performance within the AR model family while establishing itself as a formidable competitor among diverse generative methods across different paradigms.

**AR Model Comparison.** NFIG achieves the best gFID (2.81) and IS (332.42) scores, significantly outperforming other AR models. Compared to VAR-16 (gFID: 3.55, IS: 274.4), our approach reduces FID by 0.74 and improves IS by more than 21%. XQGAN-310M has a better image tokenizer with rFID 0.78 with gFID 2.96. This indicates that NFIG's performance improvement is not solely due to a good image tokenizer, but more importantly, to the injection of image frequency priors. Additionally, the proposed approach outperforms VAR-d20, a relatively larger model, while delivering 25% faster inference speed.

**Cross-family Comparison.** NFIG achieves competitive performance with the best models from other families. NFIG outperforms the best mask diffusion model RCG, which has a gFID score of 3.49 and IS score of 215.5. While some GANs like StyleGAN-XL have lower gFID scores (2.30) with moderate IS (265.1), and diffusion models like DiT-L/2 show excellent gFID (2.27) and strong IS (278.2), NFIG uniquely balances both metrics at high levels (gFID: 2.81, IS: 332.42). This establishes NFIG as not only the leading AR model but also a strong competitor across all model types.

**Qualitative Results.** Figure 3 qualitatively shows NFIG's impressive ability to generate diverse ImageNet $256 \times 256$ images across a wide variety of categories. Appendix B.5 show the Failure case of NFIG.

**Scaling Up**. To validate NFIG's scaling behavior, we train 310M and 600M parameter models for 55 epochs under computational constraints. The results are shown in Table 3.

**Performance of Different Epochs**. VAR sets different training epochs for models of different sizes: 200 epochs for 310M, 250 epochs for 600M, 300 epochs for 1B, and 350 epochs for 2B. Limited by computational resources, we focus on training our 310M model. As Table 4 shows, NFIG outperforms VAR at matched epochs and demonstrates superior parameter efficiency. At 200 epochs, NFIG already outperforms VAR-d16 of the same size. With extended training, NFIG-310M achieves performance comparable to or better than VAR-d20-600M (twice the parameters) while using significantly fewer resources.

Table 3: Performance of NFIG with different parameters at 55 epochs.

| Model | FID↓ | IS↑ | Precision↑ | Recall↑ | Epoch | Para | steps | Time |
|---|---|---|---|---|---|---|---|---|
| NFIG-310M | 5.47 | 224.20 | **0.7569** | 0.4914 | 55 | 310M | 10 | 1 |
| NFIG-600M | **5.07** | **225.16** | 0.7184 | **0.5546** | 55 | 600M | 10 | 1.25 |

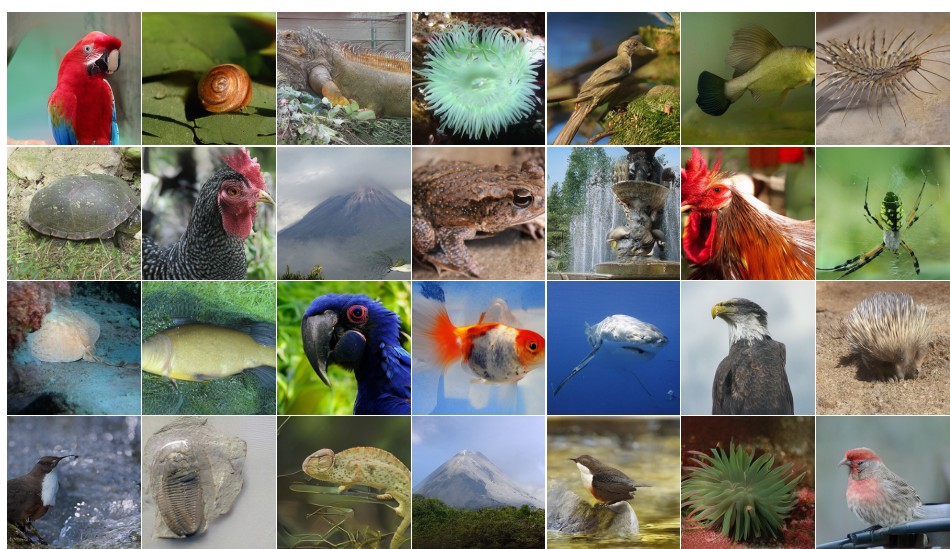

Figure 3: Generated $256 \times 256$ examples by NFIG trained on Imagenet.

Table 4: The performance of NFIG and VAR at different epochs.

| Model | Epochs | FID↓ | IS↑ | Pre↑ | Rec↑ | Params | Steps | Time |
|---|---|---|---|---|---|---|---|---|
| VAR-d16 | 200 | 3.55 | 274.4 | **0.84** | 0.51 | 310M | 10 | 1 |
| VAR-d20 | 250 | 2.95 | 302.6 | 0.83 | 0.56 | 600M | 10 | 1.25 |
| NFIG(ours) | 200 | 3.35 | 309.2 | 0.79 | 0.55 | 310M | 10 | 1 |
| NFIG(ours) | 250 | 3.16 | 311.9 | 0.78 | 0.56 | 310M | 10 | 1 |
| NFIG(ours) | 300 | 2.93 | 325.9 | 0.79 | 0.56 | 310M | 10 | 1 |
| NFIG(ours) | 350 | **2.81** | **332.4** | 0.77 | **0.59** | 310M | 10 | 1 |

## 4.3 Ablation Study

To evaluate the contribution of various components within our proposed NFIG model, we perform a comprehensive ablation analysis on the ImageNet validation set. Table 5 summarizes the results of

this study, with performance evaluated using rFID and gFID. We start with the baseline AR model with a sequence length of 256, which achieves an rFID of 1.62 and a gFID of 18.65.

**Image Tokenizer.** We incrementally add components to progressively improve the model's performance. First, incorporating frequency-guided residual quantization into the VAR framework reduces the rFID to 1.40. Next, we integrate the DINO discriminator from VAR's tokenizer, which substantially improves the rFID from 1.40 to 0.85.

**Transformer.** We then utilize the image tokenizer (FR-VAE) to train the transformer model. Without Top_k and Classifier Free Guidance (CFG), NFIG achieves a gFID of 9.7. The addition of Top_k sampling strategy further reduces the gFID to 6.83. Finally, incorporating CFG yields the best overall performance, maintaining the rFID at 0.85 while dramatically improving the gFID to 2.81.

Our experimental results demonstrate that the combination of FR-VAE with CFG provides optimal generation quality. Moreover, we observe that both DINO discriminator and FR-VAE contributes significantly to improving rFID for the image tokenizer. Additionally, Top_k sampling and CFG prove essential for reducing gFID. These results underscore the significance of discriminator guidance and conditional generation strategies for improving image generation quality.

### 4.4 Motivation Verification

**Frequency Distribution Analysis.** The experimental results in Figure 4 demonstrate the progressive refinement of generated images and the effective capture and synthesis of multi-scale visual features by FR-VAE. The frequency spectrum visualizations reveal the model's ability to hierarchically incorporate information from low to high frequencies, resulting in generated images with rich details and natural appearance. Appendix B.2 compares the frequency keep ability of NFIG and VAR. These

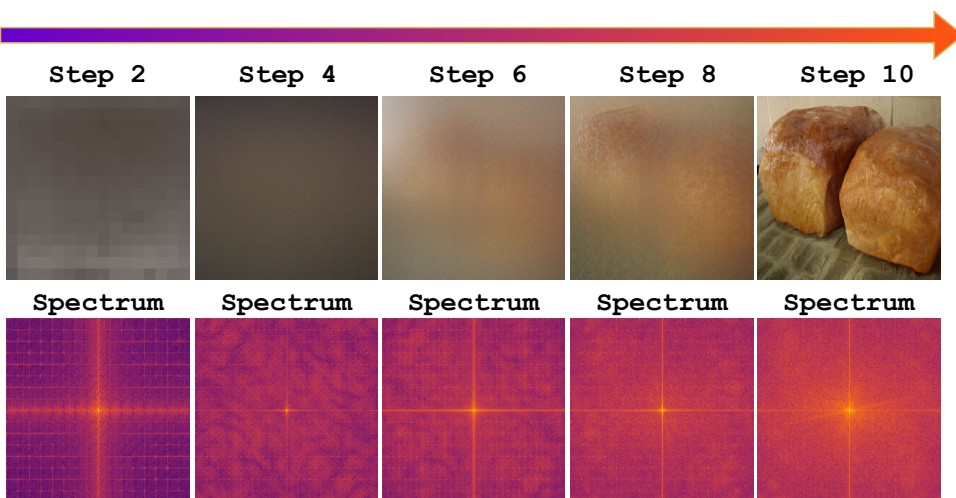

Figure 4: Generated images at different steps 2, 4, 6, 8, 10 of a 10-step process by FR-VAE, with corresponding frequency spectrum. In these spectrograms, brightness (red/yellow) indicates higher frequency energy while darker colors (blue) represent lower energy components. The center of each plot shows low-frequency information, with frequencies increasing radially outward, revealing the evolving distribution during the generation process.

results demonstrate that NFIG's frequency-guided approach enables more effective feature learning, particularly at lower resolutions, by maintaining balanced loss values throughout the hierarchical generation process.

**Frequency Guidance.** Similar to NFIG, VAR follows a "coarse-to-fine" approach but differs significantly in loss computation across resolutions. VAR computes loss between different resolutions and the raw image, causing disproportionately large loss values at lower resolutions. In contrast, NFIG utilizes frequency components to guide feature learning, providing more balanced loss values throughout generation. As Figure 5 shows, this leads to dramatic variations in vector quantization

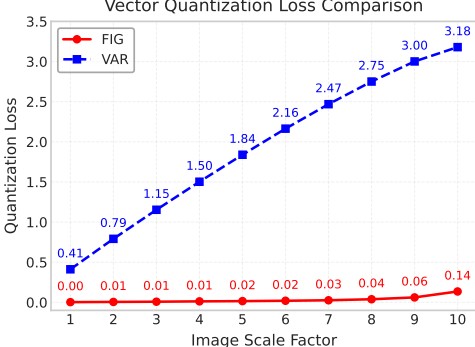

Figure 5: Vector quantization loss comparison between NFIG and VAR across image scales.

Table 5: Ablation study on the improvement of NFIG. We evaluate rFID and gFID on the ImageNet validation set. "FR-Quantizer" is the quantizer of FR-VAE. "DINO-Disc" means "DINO Discriminator", which denotes the discriminator used in VAR's ([19]) tokenizer.

| Type | Method | Length | Metric | |
|------|--------|--------|--------|--------|
| | | | rFID↓ | gFID↓ |
| 1 | AR | 256 | 1.62 | 18.65 |
| *Image Tokenizer* | | | | |
| 2 | + FR-Quantizer | 680 | 1.40 | – |
| 3 | + DINO-Disc | 680 | 0.85 | – |
| *Generation Transformer* | | | | |
| 4 | + AdaLN | 680 | 0.85 | 9.7 |
| 5 | + Top_k | 680 | 0.85 | 6.83 |
| 6 | + CFG | 680 | 0.85 | 2.81 |

loss—VAR exhibits substantially higher values across all scale factors, while NFIG maintains considerably lower loss values across all resolutions.

## 5 Conclusion

This paper introduces Next-Frequency Image Generation, a novel autoregressive framework that decomposes image generation into frequency-guided stages. Our key insight leverages the hierarchical spectral distribution of natural images: low-frequency components encode global structures and long-range dependencies, while high-frequency components contain local details requiring greater information entropy. By progressively generating from low to high frequencies, the proposed method significantly outperforms existing models with comparable parameter counts, demonstrating superior quality metrics while maintaining computational efficiency. Experiments confirm that our frequency-guided approach represents an important advancement in autoregressive image synthesis.

## 6 Limitation and Future Work

Our frequency-guided autoregressive image generation approach shows promise, but has limitations. Improving frequency decomposition. A primary issue is the simplistic frequency band division by scale, which inadequately captures information in the first band. Implementing a more rigorous division based on statistical analysis and physical principles would likely enhance NFIG's performance. Recent advances in beneficial noise theory [45, 46] suggest that properly designed noise can reduce task complexity, which could inform better frequency decomposition and data augmentation strategies [47, 48]. Extension to other modalities. Beyond 2D spatial frequency, future work could extend to video generation by incorporating temporal frequency decomposition, or to 3D object generation where frequency analysis is vital for accurate light field representation. For multi-modal generation, techniques that enhance cross-modal alignment through learnable noise [49] may offer insights for frequency-based fusion strategies. Privacy-preserving generation. Adversarial noise techniques [50] could be integrated with our framework to ensure privacy protection in generated content. Due to computational constraints and time limitations, these promising directions remain unexplored and are left for future investigation.

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

## Appendix A: Fourier Analysis in Natural Images

This appendix provides essential mathematical formulations and conceptual insights into the Fourier analysis of natural images, building upon the concepts discussed in the main text. We focus on the Discrete Fourier Transform (DFT) and its implications for image representation and the characteristics of natural scenes.

### A.1 2D Discrete Fourier Transform (2D DFT)

The 2D DFT transforms an $M \times N$ digital image $f(x, y)$ from the spatial domain (where $x \in \{0, \ldots, M-1\}$ and $y \in \{0, \ldots, N-1\}$ are spatial coordinates) to the frequency domain, yielding an $M \times N$ representation $F(u, v)$ (where $u \in \{0, \ldots, M-1\}$ and $v \in \{0, \ldots, N-1\}$ are frequency coordinates). The formula is given by:

$$F(u, v) = \sum_{x=0}^{M-1} \sum_{y=0}^{N-1} f(x, y) e^{-j2\pi\left(\frac{ux}{M} + \frac{vy}{N}\right)} \tag{8}$$

Here, $j$ is the imaginary unit ($j^2 = -1$), and the exponential term represents the basis functions (complex sinusoids) at different frequencies $(u, v)$.

### A.2 Inverse 2D Discrete Fourier Transform (2D IDFT)

The 2D IDFT allows us to reconstruct the original spatial domain image $f(x, y)$ from its frequency domain representation $F(u, v)$. The formula is:

$$f(x, y) = \frac{1}{MN} \sum_{u=0}^{M-1} \sum_{v=0}^{N-1} F(u, v) e^{j2\pi\left(\frac{ux}{M} + \frac{vy}{N}\right)} \tag{9}$$

Note the scaling factor $\frac{1}{MN}$ and the positive sign in the exponent compared to the forward transform.

### A.3 Magnitude Spectrum and Power Spectrum

The frequency domain representation $F(u, v)$ obtained from the DFT is generally a complex number. Its magnitude, $|F(u, v)|$, is known as the **Magnitude Spectrum**, which quantifies the amplitude of each frequency component present in the image.

$$|F(u, v)| = \sqrt{\text{Re}(F(u, v))^2 + \text{Im}(F(u, v))^2}$$

Closely related is the **Power Spectrum** (or Power Spectral Density), defined as the square of the magnitude spectrum. It represents how the total signal energy is distributed across the different frequencies:

$$P(u, v) = |F(u, v)|^2$$

A key characteristic of natural images is that their power spectrum typically exhibits a rapid decay as frequency increases. Specifically, the power $P(u, v)$ tends to fall off with increasing radial frequency $f_r = \sqrt{u^2 + v^2}$, often approximated by a $1/f_r^\alpha$ law, where $\alpha$ is a constant typically around 2. This **1/f property** implies that low spatial frequencies (corresponding to coarse structures and overall variations) contain significantly more energy than high spatial frequencies (corresponding to fine details and sharp transitions). This fundamental statistical feature of natural images is widely utilized and modeled in various image processing and computer vision tasks.

## Appendix B: Addition Experiments

VAR sets different training epochs for models of different sizes: 200 epochs for 310M, 250 epochs for 600M, 300 epochs for 1B, and 350 epochs for 2B. Limited by computational resources, we focus on

training our 310M model. As the table above shows, **NFIG outperforms VAR at matched epochs and demonstrates superior parameter efficiency**. At 200 epochs, NFIG already outperforms VAR-d16 of the same size. With extended training, NFIG-310M achieves performance comparable to or better than VAR-d20-600M (twice the parameters) while using significantly fewer resources.

## B.1 Details of Loss function

We will provide detailed mathematical formulations of our loss function components and their respective roles in the training process. The total loss function for the FR-VAE (image tokenizer for NFIG) is defined as:

$$\mathcal{L} = ||I - \hat{I}||_2^2 + ||\hat{f} - \hat{f}||_2^2 + \mathcal{L}_p(I) + 0.5\mathcal{L}_g(I). \tag{10}$$

Here, the first two terms represent the reconstruction loss for the image ($I$ vs $\hat{I}$) and and its frequency-guided quantized loss ($\hat{f}$ vs $\hat{f}$), respectively, ensuring fidelity in both pixel and feature. $\mathcal{L}_p$ is LPIPS perceptual loss and $\mathcal{L}_g$ is gan loss.

For the NFIG Transformer, which predicts the frequency tokens, we utilize a standard cross-entropy loss:

$$\mathcal{L}(T, \tilde{T}) = -\sum_{i=1}^{n} t_i \log(\tilde{t}_i) \tag{11}$$

This loss is computed between the predicted tokens $\tilde{T}$ and FR-VAE ground truth tokens $T$, ensuring accurate prediction of the quantized frequency representations across all scales.

## B.2 Frequency Keep Ability

We are grateful for your encouragement to discuss both successes and challenges. Your suggestion for a frequency analysis was particularly insightful. As you requested, we conducted a frequency-domain comparison between our model (NFIG) and VAR-16.

As requested, we provide frequency-domain comparisons between VAR-16 and NFIG: (1) **Power Spectral Density (PSD)**: Overall frequency fidelity; (2) **Frequency Keep Score (FKS)**: Weighted similarity across High/Mid/Low frequency bands (weights: 0.15, 0.28, 0.57, emphasizing structural low-frequency information).

Our analysis revealed that while both models effectively preserve low-frequency information, and **NFIG preserves middle and high frequency information with higher fidelity**.

| Model | PSD↓ | FKS↑ | Low↑ | Middle↑ | High↑ |
|---|---|---|---|---|---|
| VAR-16 | 0.87 | 79.5% | 98.3% | 57.6% | 48.2% |
| NFIG(ours) | 0.47 | 87.6% | 98.9% | 75.3% | 66.7% |

## B.4 Diverse Image Types

To demonstrate broader applicability, we conducted preliminary reconstruction evaluations (FID) of FR-VAE across diverse image types.

| Model | DTD | QRCODE | Diagrams | Chest-X | CelebA-HQ | COCO | LSUN-Bedroom |
|---|---|---|---|---|---|---|---|
| FR-VAE | 6.86 | 11.01 | 21 | 0.74 | 3.51 | 7.51 | 6.12 |

## B.5 Failure Case

Despite strong overall performance, NFIG occasionally produces visual artifacts. As shown in Figure 6, these include anatomical errors (extra bird leg), texture abnormalities (goldfish patterns), and fine

detail loss (bird claws). Red boxes highlight the anomalies. These failures reflect challenges in maintaining semantic consistency across frequency bands. The issues are particularly pronounced for complex structures and fine details. Such limitations are common to frequency-based generation approaches and present opportunities for future improvement.

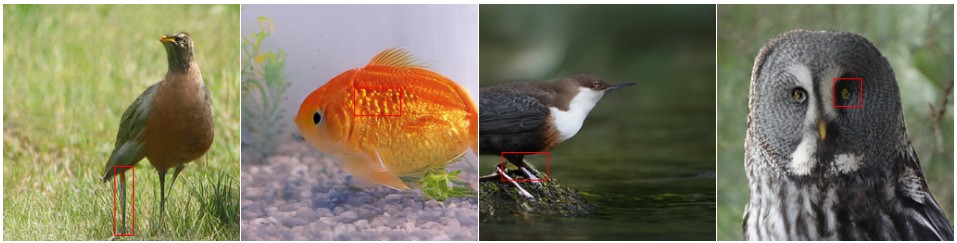

Figure 6: Failure case for NFIG.

