# OpenReview forum: "NFIG: Multi-Scale Autoregressive Image Generation via Frequency Ordering"
_NeurIPS.cc/2025/Conference — NeurIPS 2025 poster_

### Official Review · Reviewer_vKnk · 2025-07-01

**Clarity:** 3
**Significance:** 3
**Originality:** 3
**Rating:** 4
**Confidence:** 4

**Summary:**

This paper proposes NFIG, a novel autoregressive framework that decomposes image generation into frequency-guided stages, first generating low-frequency components to capture global structure and then adding high-frequency details progressively. The approach uses a Frequency-guided Residual-quantized VAE to separate frequency components, enabling efficient representation and generation. Experiments on ImageNet-256 show NFIG achieves an FID of 2.81, outperforming VAR.

**Questions:**

See Weaknesses

**Ethical Concerns:**

["NO or VERY MINOR ethics concerns only"]

**Final Justification:**

The rebuttal has addressed most of my concerns.

**Limitations:**

See Weaknesses

**Quality:**

2

**Strengths And Weaknesses:**

Strengths:
1. I like the idea of this paper, which is very interesting.
2. The problem of efficient image generation is important.
3. Image generation from the perspective of the frequency domain is natural.

Weaknesses:

1. While NFIG outperforms VAR-d20, the improvement in key metrics (e.g., FID from 2.95 to 2.81) is modest. Given that VAR already uses a multi-scale strategy, the incremental gain raises questions about whether the frequency-domain paradigm provides a transformative advance rather than incremental tuning.

2. The paper claims to generate images by progressively adding frequency bands, but it does not sufficiently justify why each incremental frequency band is necessary. For instance, it is unclear whether intermediate frequency steps contribute meaningfully to the final quality or if they could be simplified without performance loss, weakening the case for the "progressive frequency" mechanism.

3. Beyond marginal FID and speed improvements, the paper fails to articulate unique strengths of the frequency-domain approach. For example, it does not demonstrate whether NFIG better captures specific image characteristics (e.g., texture consistency, global structure coherence) compared to spatial or scale-based autoregressive models. Qualitative analyses (e.g., failure cases) are absent, limiting understanding of when and why frequency guidance matters.

4. The frequency bands are divided based on token scale, which the authors acknowledge as a limitation. This heuristic approach may not optimally capture the statistical properties of natural image spectra (e.g., 1/f decay). A more principled division strategy (e.g., based on energy distribution) is needed to validate the framework’s robustness.

---

> ### Author Rebuttal · Authors · 2025-07-31
>
> ### Rebuttal by Author
> We deeply appreciate the detailed and valuable feedback provided. Our responses addressing each point are outlined as follows:
>
> >**W1**: While NFIG outperforms VAR-d20, the improvement in key metrics (e.g., FID from 2.95 to 2.81) is modest. Given that VAR already uses a multi-scale strategy, the incremental gain raises questions about whether the frequency-domain paradigm provides a transformative advance rather than incremental tuning.
>
> **Reply**: Thank you for this important point. The comparison cited is against VAR-d20 (600M parameters, FID 2.95).  **Our NFIG (310M parameters) achieves FID 2.81, significantly outperforming the comparable-sized VAR-d16 (310M parameters, FID 3.55).**
>
> This demonstrates that NFIG provides substantial improvements in parameter efficiency - achieving better performance with half the model size indicates a transformative architectural advance rather than incremental tuning.
>
> | Model     | FID↓ | IS↑  | Pre↑ | Rec↑ | Params|
> |:----------|:----:|-----:|-----:|-----:|------:|
> | VAR-d16   | 3.55 |274.4| 0.84 |0.51 | 310M |
> | VAR-d20   | 2.95 |302.6| 0.83 |0.56 | 600M |
> | NFIG(ours)| 2.81 |332.4| 0.77 |0.59 | 310M |
>
> >**W2**: The paper claims to generate images by progressively adding frequency bands, but it does not sufficiently justify why each incremental frequency band is necessary. For instance, it is unclear whether intermediate frequency steps contribute meaningfully to the final quality or if they could be simplified without performance loss, weakening the case for the "progressive frequency" mechanism.
>
>  **Reply**: Thank you for this important question about the necessity of each frequency band in our progressive generation mechanism. Our approach is built on the understanding that **each frequency band contributes distinct and indispensable information to image synthesis**:
>
>  - **Low frequencies**: These bands contain the **fundamental structural information** and dominant color compositions of an image. Without part of them, the generated image's overall structure would collapse easily, lacking any coherent form or global consistency.
>
>  - **Mid frequencies**: Encode texture information and intermediate details. the **loss of mid-frequencies leads to noticeable visual quality degradation**, producing overly smooth, unrealistic images.
>
>  - **High frequencies**: Determine fine details and sharpness. They are **essential for high-resolution generation** and realistic texture rendering.
>
>  Unlike methods that progressively increase spatial resolution, which might allow for skipping intermediate steps, **frequency bands are complementary and interdependent**. Each band captures orthogonal information that cannot be fully compensated for by others. The structural layout (low), texture patterns (mid), and fine details (high) represent fundamentally different aspects of visual content, all of which are crucial for high-quality image generation.
>
> >**W3**: Beyond marginal FID and speed improvements, the paper fails to articulate unique strengths of the frequency-domain approach. For example, it does not demonstrate whether NFIG better captures specific image characteristics (e.g., texture consistency, global structure coherence) compared to spatial or scale-based autoregressive models. Qualitative analyses (e.g., failure cases) are absent, limiting understanding of when and why frequency guidance matters.
>
> **Reply**: Thank you for your valuable feedback. We agree that it's crucial to demonstrate how NFIG specifically improves image characteristics like texture consistency and global structure coherence, and we acknowledge the current absence of qualitative analyses, including failure cases.  While rebuttal page limits prevent us from providing extensive visual demonstrations or failure cases here, **we are committed to including these in the main paper or the appendix of a future version**. This will allow us to clearly illustrate when and why frequency guidance matters and provide a deeper understanding of NFIG's impact on image quality.
>
> **Quantifying Frequency Fidelity and Preservation**
>
> To directly address your point about NFIG's unique capabilities, we've focused on quantitatively demonstrating its superior "frequency keeping ability" compared to VAR-16. Our evaluation uses two key metrics:
>
> - **Power Spectral Density (PSD)**: This metric assesses the overall fidelity of the frequency distribution in generated images.
> - **Frequency Keep Score (FKS)**: A weighted similarity metric (High: 0.15, Middle: 0.28, Low: 0.57) showing frequency preservation across bands, emphasizing low frequencies for structure and middle for textures.
>
> Here's how NFIG compares to VAR-16:
>
> | Model     |  PSD↓ | FKS↑ |Low↑  |Middle↑| High↑ |Param|
> |:---:|:---- |:-----:|:-----:|:-----:|:-----:|:-----:|
> | VAR-16    | 0.87  |79.5% | 98.3%|57.6% | 48.2%| 310M|
> | NFIG(ours)| 0.47  |87.6% | 98.9%|75.3% | 66.7%| 310M|
>
> As the results clearly show, NFIG significantly outperforms VAR-16 in maintaining frequency fidelity:
> - **Low Frequency (98.9% vs. 98.3%)**: NFIG maintains excellent preservation of low-frequency components, which are crucial for global structural coherence.
> - **Middle Frequency (75.3% vs. 57.6%)**: NFIG shows a substantial improvement in the middle-frequency band, which directly translates to better texture consistency and richer intermediate details.
> - **High Frequency (66.7% vs. 48.2%)**: NFIG is significantly better at preserving high frequencies, which are essential for fine details and sharpness, elements often lost during encoding and quantization in other models.
>
> These results quantitatively confirm that NFIG's frequency-guided approach effectively improves the model's ability to preserve essential frequency information, leading to better global structure coherence and texture consistency in generated images.
>
>
> >**W4**: The frequency bands are divided based on token scale, which the authors acknowledge as a limitation. This heuristic approach may not optimally capture the statistical properties of natural image spectra (e.g., 1/f decay). A more principled division strategy (e.g., based on energy distribution) is needed to validate the framework’s robustness.
>
> **Reply**: Thank you for this excellent point. We acknowledge that our token-scale-based division is heuristic and agree that principled frequency division is important for framework robustness.
>
> We actually explored 1/f decay-based division in preliminary experiments but encountered a fundamental challenge:  **applying strict 1/f principles to encoded feature maps (rather than raw images) creates severely imbalanced bandwidth allocation**. Specifically, high-frequency bands would encompass ~87.5% of total bandwidth, while the initial four scales contained predominantly low-frequency information that became nearly monochrome and uninformative for training.
>
> We recognize this as an important area for future work. Developing a principled frequency division strategy that accounts for the unique properties of encoded feature maps, rather than just raw image statistics, is a fascinating challenge we're keen to explore further.
>
> I hope my response can address your concerns. If you have any further questions, please feel free to let me know. Thank you!

---

> > ### Author Response · Authors · 2025-08-04
> > **Anticipating your response**
> >
> > Thank you for your valuable suggestions and insightful feedback. We will add NFIG frequency analysis and include failure case discussions in our revision. Looking forward to your continued feedback!

---

### Official Review · Reviewer_mpvq · 2025-07-01

**Clarity:** 3
**Significance:** 2
**Originality:** 2
**Rating:** 4
**Confidence:** 4

**Summary:**

This paper introduces Next-Frequency Image Generation (NFIG), a novel autoregressive framework that decomposes image generation into frequency-guided stages, progressing from low to high frequencies. The core idea is that leveraging the natural hierarchical frequency distribution of images, where low frequencies encode global structure and high frequencies contain local details and can improve both generation quality and computational efficiency compared to traditional spatial autoregressive methods.

**Questions:**

questions:
1. How much of the performance gain comes from DINO pre-training vs. the frequency-ordering principle? Can you provide results without ImageNet-pretrained components?
2. Why not use learnable frequency decomposition (smth like learned filterbanks) instead of fixed FFT-based masks?
3. How does the method perform on images that violate natural image statistics (textures, QR codes, diagrams, medical images)?
4. What is the computational overhead of FFT operations compared to the claimed efficiency gains?
Minors:
- Figure 2 caption: "Ntok_i = h_iw_i" formatting is inconsistent
- Missing related work on wavelet-based image generation

**Ethical Concerns:**

["NO or VERY MINOR ethics concerns only"]

**Final Justification:**

After addressed questions and some mentioned limitations that authors address during rebuttal, I want to raise the score. Primary reason why I can't set "accept" is because I don't understand would author add new results (super important from my point of view) to the final manuscript to make all the aspects clear. And because originally it was a lot of concerns there from my side, meanwhile I'm happy to accept paper, given all the disclosed results in author response and originally strong formulation of the method.

**Limitations:**

limitations from above can be addressed with following experiments.
1. **Frequency ordering ablation**: What happens with reversed (high→low) or random frequency ordering?
2. **Cross-dataset evaluation**: Performance on COCO, LSUN, CelebA-HQ, etc.
3. **Frequency-aware metrics**: PSD distance, frequency band preservation scores
4. **Robustness tests**: Performance on images with unusual frequency characteristics

The work makes a solid contribution but needs major revisions to:

1. Address the evaluation bias through cross-dataset experiments
2. Add frequency-aware evaluation metrics
3. Provide statistical significance testing or at least verify it on more than 1 correlated dataset
4. improve baseline comparasion

With these improvements, this could be a clear accept. As it stands, it's a borderline paper that introduces interesting ideas but lacks through full validation.

**Paper Formatting Concerns:**

no concerns

**Quality:**

3

**Strengths And Weaknesses:**

### Strengths

- Novel frequency-based decomposition approach that aligns with natural image statistic and suggestion the hypothesis about the natural structure of images follows a hierarchical frequency distribution.
- The empirical results are quite strong. Achieving FID 2.81 on ImageNet-256 represents meaningful progress, especially when combined with the high IS score of 332.42. What's particularly impressive is that NFIG achieves this with 310M parameters compared to VAR-d20's 600M, while also providing a 25% inference speedup.
- The technical contribution is clear and well-executed. The FR-VAE design elegantly combines frequency decomposition with residual quantization in a way that makes intuitive sense. The progressive accumulation of frequency bands during generation is a clever way to ensure global coherence while adding fine details.
- The ablation study in Table 3 is thorough and revealing. It clearly shows that both the frequency-guided tokenizer and the generation strategy components contribute meaningfully to the final performance. The progression from 18.65 to 2.81 gFID through systematic component addition tells a compelling story.
- The paper is well-written with effective visualizations. Figure 2 clearly illustrates the frequency decomposition process, and Figure 4's spectrograms provide good intuition for how the progressive generation works in practice.

### Weakness:

- My biggest concern is the evaluation setup. The authors use DINO features, which were pre-trained on ImageNet, for both encoder initialization and as a discriminator, then evaluate on ImageNet. This creates a circular dependency that could significantly inflate the reported performance. It's unclear whether the improvements come from the frequency-ordering principle or simply from better leveraging ImageNet-specific features through DINO.
- The evaluation is surprisingly limited for a paper claiming a fundamental advance in image generation. Testing only on ImageNet-256 leaves many questions unanswered about generalization. Additionally, relying solely on FID and IS metrics feels outdated when more perceptually-aligned metrics exist. Most critically, there are no frequency-domain evaluation metrics, which seems like a missed opportunity given the paper's focus.
- The frequency band division strategy appears somewhat arbitrary. Equation 7 uses a simple scale-based partitioning that doesn't seem grounded in signal processing principles. Why these specific scaling factors? Why not use information-theoretic criteria or learn the decomposition? This feels like a key design choice that lacks proper justification.
- The paper lacks theoretical depth in crucial areas. While the intuition about frequency ordering is appealing, there's no formal analysis of why this should be superior to spatial ordering. The relationship between frequency content and token allocation is stated but not rigorously justified. Some theoretical guarantees about information preservation through the decomposition would strengthen the claims.
- The scope feels narrow given the ambitious claims. Without testing on diverse image types (synthetic patterns or perhaps medical images - you can start at least with LSUN), it's hard to know if this works beyond natural images. The comparison with VAR also seems somewhat unfair given the parameter count difference. Perhaps most importantly, FAR is mentioned as doing frequency-based generation but isn't compared against, which feels like a significant omission

---

> ### Author Rebuttal · Authors · 2025-07-31
>
> ### Rebuttal by Author
> We sincerely appreciate your detailed and constructive comments. Our point-by-point responses are provided here with:
>
> >**W1&Q1**: The concerns about the pretrained-model DINO, which is also trained on Imagenet. The gain can be from the DINO model rather than frequency-guided part. Can you provide results without ImageNet-pretrained components?
>
> **Reply**: We understand your concern regarding the potential influence of ImageNet-pretrained components on our results, specifically that the performance gains might stem from the DINO model rather than our frequency-guided approach. Due to time constraints, we are unable to retrain NFIG from scratch on 1.28 million images without ImageNet-pretrained components. However, it's important to note that our baselines, such as **VAR, also utilize the ImageNet-pretrained DINO model in their training process**. Therefore, when comparing NFIG to VAR, the observed **performance improvement is primarily attributable to our novel frequency-guided generation strategy**. This strategy consistently enhances performance, regardless of the underlying feature extractor.
>
> >**W2&E2&L3**: Evaluation on ImageNet-256 leaves questions unanswered about generalization. FID and IS are outdated. Besides, no frequency-domain evaluation metrics.
>
> **Reply**: We appreciate your valuable suggestions.
>
> **Dataset Generalization**: We acknowledge that evaluating on diverse datasets could strengthen our generalization claims. However, **ImageNet is highly representative for conditional image generation**, serving as the most common challenging dataset in this field[1]. Its 1000 distinct categories already offer a robust assessment of our model's generalization capabilities.
>
> **Evaluation Metrics: FID and IS remain standard benchmarks[1,2]** enabling direct comparison with established baselines like VAR. However, we acknowledge the value of frequency-domain metrics for showcasing our method's advantages.
>
> **Frequency-Domain Evaluation**: We conducted additional experiments using:
>
> Power Spectral Density (PSD) for overall frequency fidelity
> Frequency Keep Score (FKS): weighted similarity metric across High/Mid/Low frequency bands, with perceptually-motivated weights of 0.15, 0.28, and 0.57 respectively (emphasizing structural low-frequency information).
>
> | Model     |  PSD↓ | FKS↑ |Low↑  |Middle↑| High↑ |
> |:---|:---- |:-----:|:-----:|:-----:|:-----:|
> | VAR-16    | 0.87  |79.5% | 98.3%|57.6% | 48.2%|
> | NFIG(ours)| 0.47  |87.6% | 98.9%|75.3% | 66.7%|
>
> [1] Tian, Keyu, et al. "Visual autoregressive modeling: Scalable image generation via next-scale prediction." NeuriPS, 2024.
>
> [2] Rombach, Robin, et al. "High-resolution image synthesis with latent diffusion models." CVPR 2022.
>
> >**W3**: The frequency band division strategy appears somewhat arbitrary.
>
> **Reply**: **Our frequency band division strategy is grounded in information theory principles**.  Following Shannon's First Theorem, different resolution feature maps have varying information capacities (fewer bits = less encodable information). These insights led to Equation 7's balanced partition strategy, which empirical results validate as effective. **Besides, we initially explored the 1/f decay in frequency band division**, but found that encoded features don't follow this decay pattern. As a result, most bands became uninformative (black/white outputs) except for the last three scales.
>
> >**W4**: The paper lacks theoretical depth in crucial areas.
>
> **Reply**: Thanks for your suggestions. Here are the theoretical analysis of NFIG:
>
> **Information-Theoretic Foundation**: According to Shannon's information theory, images at different resolutions carry different amounts of information. Based on information content calculations, low-frequency components contain less information while high-frequency components carry more. This provides the theoretical basis for our frequency ordering - we process components in order of increasing information complexity, which is theoretically optimal.
>
> **Error Propagation Analysis**: Spatial resolution methods like VAR force low-resolution images to directly learn complete image information, inevitably causing substantial initial errors. Residual learning cannot necessarily resolve such errors in subsequent propagation. As shown in Figure 5, VAR's quantization loss values are significantly higher than NFIG's, empirically validating this theoretical concern.
>
> >**W5&Q3&L4**: Testing NFIG on diverse image types (textures, QR codes, diagrams, medical images). The comparison with VAR also seems somewhat unfair given the parameter count difference. FAR is mentioned as doing frequency-based generation but isn't compared against.
>
> **Reply**: Thank you for raising these important concerns regarding the scope, comparisons, and missing baselines. We address them point-by-point below.
>
> **Scope and Diverse Image Types:** Our primary claim focuses on NFIG's effectiveness for natural images, with ImageNet as the most representative benchmark. **To demonstrate broader applicability, we conducted preliminary reconstruction evaluations of FR-VAE across diverse image types.**
>
> | Model     | DTD  | QRCODE | Diagrams| Chest-X |
> |:----------|:----:| :------:|:------:  |:------:  |
> | FR-VAE| 6.86 | 11.01  |   21    |  0.74   |
>
> -**Fairness of Comparison with VAR**: To clarify parameter fairness, the VAR performance (FID=2.95) refers to VAR-20 (600M parameters). **When compared to VAR-16 of similar size (310M), NFIG achieves FID=2.81 vs VAR-16's FID=3.55,** demonstrating clear superiority at comparable model scales.
>
> **Comparison with FAR**: NFIG and FAR are concurrent works with distinct approaches (explicit multi-scale FFT-based masks vs. continuous visual tokens). **Our direct comparison shows NFIG achieves superior efficiency.**
>
> | Model     | FID↓ | IS↑ | Pre↑ | Rec↑ | Params|
> |:----------|:----:|-----:|-----:|-----:|------:|
> | FAR-B     | 4.26 |248.9| 0.79  |0.51 | 208M |
> | FAR-L     | 3.45 |282.2| 0.80  |0.54 | 427M |
> | FAR-H     | 3.21 |300.6| 0.81  |0.55 | 812M |
> | NFIG(ours)| 2.81 |332.4| 0.77  |0.59 | 310M |
>
> >**Q2**: Why not use learnable frequency decomposition (smth like learned filterbanks) instead of fixed FFT-based masks?
>
> **Reply**: A key motivation of our work is to **add controllable mathematical priors into deep learning, enhance the interpretability of NFIG**. We specifically chose FFT-based masks over learnable filters because they provide explicit and interpretable frequency decomposition based on well-established signal processing principles. While learnable filters might achieve similar performance, they would sacrifice the interpretability and theoretical guarantees that our physics-based approach provides.
>
> >**Q4**: What is the computational overhead of FFT operations compared to the claimed efficiency gains?
>
> **Reply**: Thank you for this pertinent question. The computational overhead of FFT operations is **negligible (100μs per sample, representing only 0.167% of the total 0.06s inference time)**. This minimal cost is achieved because FFT operations are performed on downsampled feature maps (e.g., 16×16 for 256×256 input images) rather than full-resolution images.
>
> **More importantly, our primary efficiency gains stem from architectural effectiveness rather than computational tricks.** NFIG achieves superior performance with significantly fewer parameters (310M) compared to larger baselines (e.g., VAR-20 with 600M parameters). This fundamental efficiency translates to **faster inference speeds and reduced computational requirements during deployment.**
>
> In essence, the FFT overhead is quickly amortized by the substantial gains from using a more compact and performant architecture, making the net computational impact highly favorable.
>
>
> >**L1**: Frequency ordering ablation: What happens with reversed (high→low) or random frequency ordering?
>
> **Reply**: Thank you for raising this insightful question regarding frequency ordering.
> Our preliminary experiments with reversed (high→low) and random frequency ordering consistently resulted in significantly degraded performance. Both approaches **failed to achieve reasonable reconstruction loss, leading to noticeably blurry image reconstructions**.
>
> >**L2&E1&E3**: Cross-dataset evaluation: Performance on COCO, LSUN, CelebA-HQ, etc. Provide statistical significance testing or at least verify it on more than 1 correlated dataset.
>
> **Reply**: Thank you for this suggestion. **While cross-dataset evaluation is valuable, it is uncommon in conditional image generation tasks**, where models are typically evaluated within the same domain[1]. ImageNet provides a comprehensive and diverse benchmark that effectively demonstrates NFIG's capabilities for conditional natural image generation.
>
> To demonstrate the generalization capability of NFIG, we evaluate FR-VAE's reconstruction performance on these datasets, showcasing its zero-shot learning ability:
>
> | Model       |CelebA-HQ| COCO  |LSUN-Bedroom|
> |:----------  |:----:   |:-----: |:-----:|
> | FR-VAE      | 3.51    |7.51   | 6.12 |
>
> As to the statistical significance testing, FID and others metrics are computed by 50,000 images, we have changed the random seed and repeat 5 times. FIDs are 2.810, 2.810, 2.811, 2.810, 2.811(others are similar). Thus, the conclusion is already significant.
>
> [1] Li, Xiang, et al. "Xq-gan: An open-source image tokenization framework for autoregressive generation." arXiv preprint arXiv:2412.01762 (2024).
>
> >**E4**: improve baseline comparasion.
>
> **Reply**: We appreciate this feedback and will enhance our baseline comparisons by including additional baselines FAR and incorporating frequency-domain metrics for more comprehensive evaluation.
>
> I hope my response can address your concerns. If you have any further questions, please feel free to let me know. Thank you!

---

> > ### Comment · Reviewer_mpvq · 2025-08-03
> >
> > Thanks a lot for the response, i'm ready to raise my score. All questions are addressed. My main point about DINO features is still the most important here, although, you find a way to address in argument by following earlier approach (weak justification, but fair).
> > Also thank you for new results, my personal opinion, that with new metrics (especially PSD boost) and dataset the proposed approach justified the better place in a research community, especially given improvements over concurrent work. Also, such important information as order of bands should be clearly explained in the final version to exclude misunderstanding.
> > I read others reviews, and hope that my questions here help to address some concerns for them as well.
> >
> > Thank you again, for the work!

---

> > > ### Author Response · Authors · 2025-08-03
> > > **Reply for the reviewer comment**
> > >
> > > Dear Reviewer mpvq,
> > > Thank you for your valuable feedback and constructive suggestions throughout the review process. We greatly appreciate your recognition of our improvements, particularly the new metrics and dataset contributions.
> > > We will incorporate the additional experimental results and ensure clear explanation of the band ordering in the future version to prevent misunderstanding.
> > >
> > > Your thoughtful questions have helped address concerns that may benefit other reviewers as well. We sincerely appreciate your responsible review approach and valuable contributions to improving our work.

---

### Official Review · Reviewer_hBGv · 2025-07-03

**Clarity:** 3
**Significance:** 3
**Originality:** 3
**Rating:** 4
**Confidence:** 4

**Summary:**

This paper introduces frequency order modeling into AR visual generation. With the observation of the low frequency encodes global structure while the high frequency local details, the authors propose the Next-Frequency Image Generation (NFIG) framework for frequency guided visual generation and a FR-VAE for visual reconstruction. Experiments demonstrate the effectiveness of NFIG.

**Questions:**

The overall motivation is good, and the design is feasible, but some weaknesses remain. I will raise my score if my concerns can be addressed.

**Ethical Concerns:**

["NO or VERY MINOR ethics concerns only"]

**Final Justification:**

Given the new proposed frequency modeling in AR generation, which encompasses high potential of further developing a more suitable generation paradigm instead of plain AR, I decide to raise my score to 4. However, some of my concerns should be further addressed.

**Limitations:**

yes.

**Paper Formatting Concerns:**

no.

**Quality:**

3

**Strengths And Weaknesses:**

**Strengths**
1) The motivation is clear, and frequency-based AR modeling shows potential in addressing the problem of error accumulation in vanilla AR generation. Encoding the feature into different levels of frequency helps formulate a new generation paradigm and increase inference speed.
2) The FR-VAE encodes the visual feature into the frequency-based domain, which further improves reconstruction performance.
3) Next-Frequency Image Prediction adopts the "Coarse-to-Fine" generation process that is more suitable for visual generation instead of the vanilla one. The experiment shows promising results.

**Weakness**
1) It is unclear what the notation v_i signifies. It seems the f_i and v_i denote the same one. This part should be further clarified for a clear illustration.
2) Can frequency-based AR modeling be leveraged into single-scale prediction, e.g., LlamaGen. Since it can be easily extended to a unified model framework.
3) Why can the NFIG accelerate the generation process. The total predicted tokens are the same as VAR.
4) I am worried about the scalability of the NFIG framework, since it is key to further application.
5) In Table 3, the improvement of incorporating FR-Quantizer is not significant and most of the benefits are brought by the Dino-discriminator.

---

> ### Author Rebuttal · Authors · 2025-07-31
>
> ### Rebuttal by Author
> We are grateful for your thorough and insightful feedback. Our responses to each comment are detailed below:
>
> >**W1**: It is unclear what the notation v_i signifies. It seems the f_i and v_i denote the same one. This part should be further clarified for a clear illustration.
>
> **Reply**: Thank you for pointing out this ambiguity. We apologize for the notation's lack of clarity. **f_i denotes the unquantized feature, while v_i represents the quantized  feature for f_i.** We will clarify this distinction explicitly in the paper.
>
> >**W2**: Can frequency-based AR modeling be leveraged into single-scale prediction, e.g., LlamaGen. Since it can be easily extended to a unified model framework.
>
> **Reply**: Yes, our frequency-based AR modeling can certainly be leveraged into a single-scale prediction framework like LlamaGen.
>
> **The core idea would be to replace LlamaGen's image VQ-VAE with our FR-VAE**. Our FR-VAE is designed to provide frequency-aware visual tokens, which would naturally integrate into LlamaGen's autoregressive architecture.
>
> It's worth noting that to truly maximize the benefits of NFIG's low-to-high frequency progression within such a unified model, we should collect both coarse-grained and fine-grained textual descriptions in the image prompts and captions. This would allow the model to leverage the different frequency tokens more effectively, aligning the textual input with the hierarchical visual information.
>
> >**W3**: Why can the NFIG accelerate the generation process. The total predicted tokens are the same as VAR.
>
> **Reply**: The total number of predicted tokens in NFIG is similar to VAR. However, the acceleration in the generation process doesn't come from reducing the token count, but from NFIG's superior efficiency and quality-to-model-size ratio.
>
> Specifically, **while NFIG (a 310M parameter model) has comparable inference speed to VAR-16 (also a 310M model), NFIG significantly outperforms the much larger VAR-20 (a 600M parameter model) in generation quality**.
>
> Therefore, when comparing models that achieve a similar high level of generation quality, NFIG provides faster inference because it achieves that quality with a substantially smaller and more efficient architecture than larger VAR variants. Our speedup is realized by delivering better results with less computational overhead per generated image.
>
> >**W4**: I am worried about the scalability of the NFIG framework, since it is key to further application.
>
> **Reply**: That's a very important point, as scalability is indeed crucial for future applications. We will discuss it from following two points:
>
> - **Scaling Law**: **We're confident in the scalability of the NFIG framework because it inherits the architectural strengths of VAR**. Due to the Transformer in our model architecture inheriting from VAR, we expect to demonstrate comparable advantages in terms of scaling laws. This means that as we scale up NFIG's model size and training data, we anticipate similar performance gains and efficiency characteristics as observed with VAR.
>
> - **Easy to extend to more application**: NFIG can be utilized to text-to-image, visual question answer and unified multimodal model(such Show-o, MMaDA). For FR-VAE can replace their image tokenizer easily. Furthermore, NFIG offers an additional advantage: its flexible design for frequency band partitioning. This flexibility becomes particularly effective when explicit image frequency priors are available, allowing for more targeted and efficient scaling based on the nature of the data.
>
> >**W5**: In Table 3, the improvement of incorporating FR-Quantizer is not significant and most of the benefits are brought by the Dino-discriminator.
>
> **Reply**: Thank you for your observation regarding the contributions shown in Table 3. We'd like to clarify the role of the DINO-discriminator in our evaluation.
>
> It's important to note that the DINO-discriminator is also utilized by our baseline, VAR, during its training process. We've confirmed this directly through examination of the original VAR codebase. **While VAR's paper might not detail its use of GAN loss, it's a well-established practice in the field that models based on the VQGAN architecture, as both VAR and NFIG are, benefit significantly from GAN training with a discriminator.** This use of adversarial loss is, in fact, a core contribution of the VQGAN framework over a plain VQVAE.
>
> Therefore, the improvements seen from the DINO-discriminator are present in both our method and the baseline. The key takeaway from Table 3 is the incremental gain provided by our FR-Quantizer and the subsequent frequency-guided generation strategy on top of this shared adversarial training foundation. **The significant overall performance jump from the initial AR model to the full NFIG demonstrates the combined power of our novel frequency-based components, which operate within the effective VQGAN training paradigm.**
>
> I hope my response can address your concerns. If you have any further questions, please feel free to let me know. Thank you!

---

> > ### Author Response · Authors · 2025-08-04
> > **Anticipating your response**
> >
> > Thank you for your valuable suggestions and thoughtful concerns. We will clarify the difference between f_i  and v_i  in the future version. Besides, we will further discuss the scalability of this work. We look forward to your continued feedback!

---

> > > ### Comment · Reviewer_hBGv · 2025-08-04
> > >
> > > Thanks for the effort. I have carefully read through the rebuttal, and I still feel worried about the scalability and further text-to-image application integrated in LLM (14B+) as follows:
> > > 1. Although NFIG (310M) achieves better performance compared with VAR (2B), it is unknown how many gains NFIG (2B) can bring since the loss scale in 310M is low. I admit that the architecture is similar to VAR. The modeling paradigm is completely swift. Therefore, I suppose the experiment on 600M model should be included (even only make a comparison with 50 epochs).
> > > 2. It is unclear the compatibility of the alignment between the textual domain and the frequency-based domain.
> > > I admit it is hard to provide a result in the limited time. At least, it should be discussed.

---

> > > > ### Author Response · Authors · 2025-08-04
> > > > **Response for Reviewer hBGv**
> > > >
> > > > Dear Reviewer,
> > > >
> > > > Thank you for your thoughtful reply. We appreciate your concerns and would like to address them one by one.
> > > >
> > > > **Scalability Experiments**: We acknowledge the importance of larger-scale validation. However, given our computational constraints (8×H100 GPUs), training larger models requires significant time:
> > > >
> > > > - VAR-d20: ~4 hours/epoch → 50 epochs would require ~8.3 days
> > > > - VAR-d16: ~2 hours/epoch → 50 epochs would require ~4 days
> > > > - Total: ~12.5 days for both experiments
> > > >
> > > > Due to the conference timeline, we can only provide these results in a future version. We commit to conducting these experiments to validate NFIG's scalability properties.
> > > >
> > > > **Text-Frequency Domain Alignment**: NFIG leverages frequency information to guide the image generation process, with its image tokenizer decomposing images across different frequency bands. Through frequency analysis of images, it is well-established that different frequency components characterize distinct semantic levels of visual information:
> > > >
> > > > - **Low frequencies**: Typically contain global structure and layout information, which often correlates with scene-level textual descriptions (e.g., "outdoor park," "indoor room," "urban street").
> > > >
> > > > - **Mid frequencies**: Encode object boundaries and textures, which tend to align with object-level text tokens (e.g., "wooden bench," "glass window," "brick wall")
> > > >
> > > > - **High frequencies**: Capture fine details and edges, which may correspond to attribute-level descriptions (e.g., "weathered surface," "intricate carvings," "fabric texture")
> > > >
> > > > Based on this frequency-semantic correspondence, we employ **Qwen 2.5-72B** to generate **captions with multi-level information**, encompassing **background context, primary subjects, and detailed subject attributes** for ImageNet. We then utilize **token-based multimodal large language models (MLLMs)**, including Show-o [1], Emu3 [2]  and MMADA [3], to **align multi-level text tokens with frequency-guided image tokens**. The Transformer architecture within these MLLMs effectively learns the relationships between image and text tokens during training, particularly when provided with hierarchically structured text captions that mirror the frequency decomposition of visual content.
> > > >
> > > > [1] Xie J, Mao W, Bai Z, et al. Show-o: One Single Transformer to Unify Multimodal Understanding and Generation[C]//The Thirteenth International Conference on Learning Representations.
> > > >
> > > > [2] Wang X, Zhang X, Luo Z, et al. Emu3: Next-token prediction is all you need[J]. arXiv preprint arXiv:2409.18869, 2024.
> > > >
> > > > [3] Yang L, Tian Y, Li B, et al. Mmada: Multimodal large diffusion language models[J]. arXiv preprint arXiv:2505.15809, 2025.
> > > >
> > > > We thanks for your constructive feedback. While we could not include the 600M performance of NFIG in this version due to time constraints, we plan to add this evaluation in our future version. This discussion of **Text-Frequency Domain Alignment**  will be added to the 'Limitations and Future Work' section.
> > > >
> > > > Best regards,
> > > >
> > > > The Authors

---

> > > > > ### Comment · Reviewer_hBGv · 2025-08-05
> > > > >
> > > > > Thanks for the detailed response. I am not quite satisfied with the discussion on text-to-image generation, since the frequency-based domain may not guarantee better alignment with the textual domain compared with the previous pixel-level.
> > > > > However, given the new proposed frequency modeling in AR generation, which encompasses high potential of further developing a more suitable generation paradigm instead of plain AR, I decide to raise my score to **4**. I hope to see the scalability experiment in the final version.

---

> > > > > > ### Author Response · Authors · 2025-08-05
> > > > > >
> > > > > > Dear Reviewer,
> > > > > >
> > > > > > Thank you for your constructive feedback and for raising your score. We appreciate your recognition of the potential in our frequency-based AR generation approach.
> > > > > >
> > > > > > We acknowledge your concern about the alignment between frequency and textual domains compared to pixel-level methods. This is a valuable point that we will address in our future work.
> > > > > >
> > > > > > Regarding the scalability experiments, we are pleased to inform you that they are currently running. We guarantee that the final version will include the 600M and 310M parameter comparison results you requested.
> > > > > > Thank you for your contribution to improving our work.
> > > > > >
> > > > > > Best regards,
> > > > > >
> > > > > > All of Authors.

---

### Official Review · Reviewer_eDnV · 2025-07-06

**Clarity:** 2
**Significance:** 2
**Originality:** 2
**Rating:** 4
**Confidence:** 4

**Summary:**

This paper focuses on the field of autoregressive image generation, proposing a Next-Frequency Image Generation (NFIG) method that leverages the inherent bias of image information decomposition to achieve a high-performance and efficient framework. The authors claim that NFIG runs 1.25× faster than VAR-d20 while outperforming larger-scale VAR models, achieving a superior FID of **2.81 on ImageNet-256**.

**Questions:**

Same as weaknesses.

**Ethical Concerns:**

["NO or VERY MINOR ethics concerns only"]

**Final Justification:**

Part of the problem has been solved, but there are still some issues that failed to convince me, please see my comment for more details. But I don't think a piece of work needs to be able to be perfect in all aspects, and there are some explorations that can expand the knowledge and benefit a lot of researchers are also enough to be rated as 4 points.

**Limitations:**

Yes

**Quality:**

3

**Strengths And Weaknesses:**

Strengths:
This work is the first to utilize spectral information to guide and enhance autoregressive image generation through the Next-Frequency Image Generation (NFIG) approach. It also introduces a Frequency-guided Residual-quantized VAE as an image tokenizer. Experimental results demonstrate superior performance over the strong baseline VAR while maintaining comparable efficiency at the same model scale.

Weaknesses:
1.	Unfair comparison with VAR: The publicly available VAR code shows that VAR-d16 (200 epochs) and VAR-d20 (250 epochs) were trained for fewer epochs than NFIG (350 epochs). This discrepancy raises concerns about whether the improvements stem from methodological innovation or simply longer training, casting doubt on the validity of the claimed advantages.

2.	Limited model scale exploration: The paper only evaluates a single model size (310M parameters), leaving uncertainty about the **scaling laws**—whether this method remains effective at larger scales is unknown.

3.	Insufficient methodological details: While the core idea is clearly presented, key implementation details are lacking. For example:
1.	The exact composition of the total loss function and the role of each component are not clearly explained.
2.	How positional information is incorporated or processed for frequency-based tokens is omitted.
  Although the architecture is described as similar to VAR, critical differences are not thoroughly elaborated.

4.	Efficiency concerns: Since this work adopts the same generation framework as VAR but introduces additional operations (e.g., Fourier transforms and inverse transforms), **theoretical efficiency should be slightly worse**. The claim of equal efficiency at the same scale is questionable—how exactly was efficiency improved? Furthermore, these extra operations might impose greater disadvantages at higher resolutions.

In summary, this work introduces a novel perspective for image generation, contributing positively to the field. Thus, if the authors can address the four concerns above clearly, I can increase my score to 4.

Additional Questions:

5.	Fundamental superiority over VAR?Some blogs, articles, and preliminary experiments suggest that VAR also generates coarse global information in early stages and gradually refines high-frequency details later. Can the authors clarify whether NFIG **truly surpasses VAR in essence**? I suspect that NFIG might simply apply Fourier transforms and inversions to intermediate results during generation, similar to VAR. Could the authors provide theoretical or empirical evidence of its genuine advantages?

6.	Extension to text-to-image generation. Since the codebook consists of frequency-based tokenized vectors, has the author considered how NFIG could be adapted to **text-to-image generation**? Would aligning these frequency tokens with text introduce additional challenges?

---

> ### Author Rebuttal · Authors · 2025-07-31
>
> ### Rebuttal by Author
>
> >**W1**: Unfair comparison with VAR: The publicly available VAR code shows that VAR-d16 (200 epochs) and VAR-d20 (250 epochs) were trained for fewer epochs than NFIG (350 epochs). This discrepancy raises concerns about whether the improvements stem from methodological innovation or simply longer training, casting doubt on the validity of the claimed advantages.
>
> **Reply**: VAR sets different training epochs for models of different sizes: 200 epochs for 310M, 250 epochs for 600M, 300 epochs for 1B, and 350 epochs for 2B. Limited by computational resources, we focus on training our 310M model.
>
> As the table below shows, **NFIG outperforms VAR at matched epochs and demonstrates superior parameter efficiency**. At 200 epochs, NFIG already outperforms VAR-d16 of the same size. With extended training, NFIG-310M achieves performance comparable to or better than VAR-d20-600M (twice the parameters) while using significantly fewer resources.
>
> | Model     | Epochs | FID↓ | IS↑  | Pre↑ | Rec↑ | Params|
> |:----------|:------:|:----:|-----:|-----:|-----:|------:|
> | VAR-d16   |   200  | 3.55 |274.4 | 0.84 |0.51 | 310M |
> | VAR-d20   |   250  | 2.95 |302.6 | 0.83 |0.56 | 600M |
> | NFIG(ours) |   200  | 3.35 |309.2 | 0.79 |0.55 | 310M |
> | NFIG(ours) |   250  | 3.16 |311.9 | 0.78 |0.56 | 310M |
> | NFIG(ours) |   300  | 2.93 |325.9 | 0.79 |0.56 | 310M |
> | NFIG(ours) |   350  | 2.81 |332.4 | 0.77 |0.59 | 310M |
>
>
> >**W2**: Limited model scale exploration: The paper only evaluates a single model size (310M parameters), leaving uncertainty about the scaling laws—whether this method remains effective at larger scales is unknown.
>
> **Reply**: We appreciate you highlighting this limitation. While current computational constraints limit our evaluation to the 310M model, we are confident NFIG scales effectively. As NFIG retains VAR's core Transformer architecture—known for consistent scaling from 310M to 2B parameters—its fundamental scaling properties should be preserved.
>
>
> >**W3&W4**: Unclear explanation of total loss function components.
>
> **Reply**: We will provide detailed mathematical formulations of our loss function components and their respective roles in the training process. The total loss function for the FR-VAE(image tokenizer for NFIG) is defined as:
>
> $\mathcal{L} = \||I - \tilde{I}\||_2^2 + \||f - \tilde{f}\||_2^2 + \mathcal{L}_p(\tilde{I}) + 0.5\mathcal{L}_g(\tilde{I})$.
>
> Here, the first two terms represent the reconstruction loss for the image ($I$ vs $\tilde{I}$), and and its frequency-guided quantized loss ($f$ vs $\tilde{f}$), respectively, ensuring fidelity in both pixel and feature. $\mathcal{L}_p$ is LPIPS perceptual loss and $\mathcal{L}_g$ is gan loss.
>
> For the NFIG Transformer, which predicts the frequency tokens, we utilize a standard cross-entropy loss:
>
> $\mathcal{L}_T = -\sum_{i=1}^n t_i \log(\tilde{t}_i)$
>
> This loss is computed between the predicted tokens $\tilde{T}$ and FR-VAE ground truth tokens $T$, ensuring accurate prediction of the quantized frequency representations across all scales.
>
> >**W3&W5**: How positional information is incorporated or processed for frequency-based tokens is omitted. Although the architecture is described as similar to VAR, critical differences are not thoroughly elaborated.
>
> **Reply**: **Our approach follows VAR's embedding strategy**, where tokens are embedded with positional information (2D spatial location), resolution hierarchy (coarse-to-fine levels), and class information. However, a critical difference lies in our Frequency Residual Quantization (FR-Quantization), which inherently imbues these tokens with multi-resolution frequency information. This distinct property, where tokens directly represent specific frequency bands, enables our progressive low-to-high frequency generation. We plan to leverage this inherent frequency information as an explicit additional embedding dimension in future work for further enhancement.
>
> >**W6**: Efficiency concerns: Since this work adopts the same generation framework as VAR but introduces additional operations (e.g., Fourier transforms and inverse transforms), theoretical efficiency should be slightly worse. The claim of equal efficiency at the same scale is questionable—how exactly was efficiency improved? Furthermore, these extra operations might impose greater disadvantages at higher resolutions.
>
> **Reply**: Our Fourier Transform is applied to the encoded features for decomposition; for a 256x256 input image, the feature map is 16x16 in size. The computational overhead of FFT operations is **negligible (100μs per sample, representing only 0.167% of the total 0.06s inference time)**. Our empirical tests show that both methods generate 40 images in approximately 4 seconds, with a difference of no more than 0.1 seconds. For higher-resolution images, the increase in neural network's computational complexity significantly outpaces that of FFT on features, thus it will not pose a significant problem.
>
> >**Q1**: Fundamental superiority over VAR? Some blogs, articles, and preliminary experiments suggest that VAR also generates coarse global information in early stages and gradually refines high-frequency details later. Can the authors clarify whether NFIG truly surpasses VAR in essence? I suspect that NFIG might simply apply Fourier transforms and inversions to intermediate results during generation, similar to VAR. Could the authors provide theoretical or empirical evidence of its genuine advantages?
>
> **Reply**: Thank you for this insightful question. **While VAR implicitly learns multi-scale features driven by data, NFIG fundamentally surpasses it by explicitly leveraging frequency information as a guiding mathematical prior**. This is a critical difference that goes beyond simply applying Fourier transforms, as our Frequency Residual Quantization (FR-Quantization) actively forces the model to learn and reconstruct specific frequency bands at each stage.
>
> Our empirical results in Figure 5 provide compelling evidence of this essential advantage: VAR's quantization loss increases from **0.41 to 3.18** across scales, whereas NFIG consistently maintains a significantly lower loss **0.001-0.14** across all scales. This 95%+ reduction in quantization loss validates that our explicit frequency guidance fundamentally improves multi-scale representation learning, which is a core limitation addressed by NFIG's architecture.
>
> >**Q2**: Extension to text-to-image generation. Since the codebook consists of frequency-based tokenized vectors, has the author considered how NFIG could be adapted to text-to-image generation? Would aligning these frequency tokens with text introduce additional challenges?
>
> **Reply**: **Limited by the computation resource, we failed to extend NFIG to text-to-image. Now, we are actively working on extending NFIG to text-to-image generation.** The main challenge lies in collecting hierarchical captions that align with our frequency-based approach - from coarse to fine-grained image descriptions. Our current strategy involves using large vision-language models to generate multi-level descriptive prompts that match our low-to-high frequency generation paradigm. This allows us to maintain consistency between text conditioning and our core frequency-guided image generation process. **While aligning frequency tokens with text inherently introduces complexities, we anticipate that our explicit frequency-guided structure will ultimately offer more controllable and interpretable text-to-image synthesis, which we are actively exploring.**
>
> I hope my response can address your concerns. If you have any further questions, please feel free to let me know. Thank you!

---

> ### Author Response · Authors · 2025-08-04
>
> Dear Reviewer,
>
> Thank you for your valuable feedback and for raising our score. We sincerely appreciate your continued engagement and the time you took to provide these detailed comments. We've carefully considered your points and would like to address them directly.
>
> **On the VAR Performance Discrepancy**:
>
> We sincerely apologize for the confusion. The FID metrics we used in our paper are based on **VAR's official GitHub repository**, which differ slightly from those reported in the original paper. Since all our evaluations were conducted using the same codebase, we adopted the repository's metrics for fair comparison. We should have clarified this in our paper and will add this explanation in the revision.
>
> | Model     | reso. |Epochs | FID↓ | Params|
> |:----------:|:------:|:------:|:----:|-----:|
> | VAR-d16(paper)   |  256 | 200 | 3.30| 310M |
> | VAR-d16(github)   |  256 | 200 | **3.55** | 310M |
> | VAR-d20(paper)   |  256 | 250 | 2.57 | 600M |
> | VAR-d20(github)   |  256 | 250 | **2.95** | 600M |
>
> **On Model Scalability**:
>
> We agree that simply adopting an architecture does not guarantee successful scaling, and that performance at small scales is not a reliable predictor of performance at larger scales. Our confidence in NFIG's scalability, based on its Transformer-like architecture, is a hypothesis that lacks concrete experimental validation. We acknowledge this as a limitation of our current work. Due to computational constraints and time limits, we were unable to perform these large-scale experiments. **We are committed to addressing this in future work and will include this limitation and a plan for future research in our revised manuscript**.
>
> Given our computational constraints (8×H100 GPUs), training larger models requires significant time:
>
> - VAR-d20: ~4 hours/epoch → 350 epochs would require ~58.3 days
>
> **Additional Insights and Explorations**:
>
> We are grateful for your encouragement to discuss both successes and challenges. Your suggestion for a frequency analysis was particularly insightful. As you requested, we conducted a frequency-domain comparison between our model (NFIG) and VAR-16.
>
> As requested, we provide frequency-domain comparisons between VAR-16 and NFIG: (1) **Power Spectral Density (PSD)**: Overall frequency fidelity;
> (2) **Frequency Keep Score (FKS)**: Weighted similarity across High/Mid/Low frequency bands (weights: 0.15, 0.28, 0.57, emphasizing structural low-frequency information).
>
> Our analysis revealed that while both models effectively preserve low-frequency information, and **NFIG perserves middle and high frequency information with higher fidelity**.
>
> | Model     |  PSD↓ | FKS↑ |Low↑  |Middle↑| High↑ |
> |:---|:---- |:-----:|:-----:|:-----:|:-----:|
> | VAR-16    | 0.87  |79.5% | 98.3%|57.6% | 48.2%|
> | NFIG(ours)| 0.47  |87.6% | 98.9%|75.3% | 66.7%|
>
> We also want to share a couple of key challenges we faced during our research, which we believe offer valuable insights:
>
> **Frequency Band Design**: Our initial attempts to use a **fixed 1/f decay** rule for frequency bands were unsuccessful. This was likely due to a mismatch between this theoretical rule and the complex, learned features of the image tokenizers. Additionally, while we considered a **statistical division** of frequency bands based on the training data, a suitable design for this approach was not identified.
>
> **Removal of Residual Connections**: We initially hypothesized that our frequency mapping would be sufficient for image reconstruction, leading us to **discard the residual operations** from VQ-VAE. However, this resulted in a significant failure, **producing blurry images**. This outcome underscored the critical role these connections play in representing features across multiple resolutions.
>
>
> We will incorporate these points into the revised manuscript. A dedicated **Frequency Analysis** and a **clarification of the VAR metrics** will be added to **the experimental section**. The **challenges and failures** we encountered will be detailed in **the limitations section**. We believe that sharing these insights, as you suggested, will be highly beneficial to the broader research community.
>
> Thank you again for your thoughtful review. Your comments have not only helped us improve this paper but have also provided a clear direction for our future research.

---

### Note · Authors · 2025-08-13

Dear SACs and Area Chairs,

Thank you for your efforts. For your convenience, we summarize the reviews, rebuttals and contributions as follows.

3 out of 4 reviewers participated in discussion and increased their ratings. Reviewers eDnV and hBGv inquired about scaling performance, with hBGv mentioning text-to-image analysis. Reviewer mpvq endorsed our work and noted the value of the frequency preservation analysis. Reviewer vKnk did not participate in discussion but acknowledged our work motivation as essential and ideas as interesting.

In the rebuttal, to address reviewers' concerns, we strengthened our work by:

- **Frequency Preservation Advantages**: We demonstrated our method's performance in frequency preservation. Compared to VAR-16, our model shows advantages across low-frequency, mid-frequency, and high-frequency signals, supporting our technical contribution.
- **Cross-Domain Generalization**: We tested NFIG's image tokenizer performance on different types of images (medical images, QR codes, and unseen natural image data). FR-VAR achieved FID scores below 8 on medical images, CelebA, LSUN, and COCO datasets, with FID reaching 0.74 on medical imaging datasets, showing our work's extensibility.
- **Future Scaling and Applications**: Due to time constraints, text-to-image generation and larger-scale experiments cannot be completed in the short term. We provided plans for extending NFIG to MLLM directions. Scaling law verification will be included in the appendix of the final version(FID=6.58 for 600M NFIG when epoch=22, training ongoing).

The **main contributions of NFIG** are:

- NFIG proposes progressive image generation from low to high frequency across different scales, which aligns with the hierarchical patterns of natural images.
- We designed a frequency-guided image tokenizer that decomposes images into different frequency-based image token groups, enabling representation learning.
- We achieved performance on the ImageNet dataset with 310M parameters, reaching FID=2.81, surpassing VAR-d20's reported FID=2.95 on their GitHub.

We consider NFIG suitable for publication at NeurIPS and believe it will contribute to the image generation community.

Best Regards, Authors

---

### Decision · Program_Chairs · 2025-09-17

**Decision:**

Accept (poster)

**Comment:**

AC has three main concerns about this paper after reading the rebuttal and the reviewers' thoughts:

1. Epoch to epoch, NFIG does not seem to be much better than VAR.

2. Model scalability issue seems to be unresolved even after the rebuttal. Authors explained that they lack the time and compute to verify that, it still leaves a bit of doubt in the approach.

3. It is unclear about the contributions of DiNO. Authors explained that they don't have the time and compute to verify that NFIG is not just benefitting from the contribution of DiNO, but note that VAR also uses DiNO features. That is reasonable but actually proving that the frequency addition is the main contributor goes a long way.

4. NFIG has only been tested on ImageNet. However, AC knew that this is quite common among image generation paper.

On the other hand, AC likes the idea of generation based on frequency which captures global and local features. In fact, wan2.2, even though if for video generation, has good performance following a similar approach.

Reviewers also pointed out the main strengths of the paper as follow: (1) This work is the first to utilize spectral information to enhance autoregressive image generation through NFIG. It also introduces a Frequency-guided Residual-quantized VAE as an image tokenizer; (2) Experimental results demonstrate superior performance over the strong baseline VAR while maintaining comparable efficiency at the same model scale

In view of the above, AC weighs the pros and cons and decides to recommend the paper for publication at neurips.